# Correlation of Road Network Structure and Urban Mobility Intensity: An Exploratory Study Using Geo-Tagged Tweets

Li Geng [1] and Ke Zhang [2,*]

1 New York City College of Technology, City University of New York, Brooklyn, NY 11201, USA; lgeng@citytech.cuny.edu
2 University of Pittsburgh, Pittsburgh, PA 15213, USA
* Correspondence: kez11@pitt.edu

**Abstract:** Urban planners have been long interested in understanding how urban structure and activities are mutually influenced. Human mobility and economic activities naturally drive the formation of road network structure and the accessibility of the latter shapes the patterns of movement flow across urban space. In this paper, we perform an exploratory study on the relationship between the street network structure and the intensity of human movement in urban areas. We focus on two cities and we utilize a dataset of geo-tagged tweets that can form a proxy to urban mobility and the corresponding street networks as obtained from OpenStreetMap. We apply three network centrality measures, including closeness, betweenness and straightness centrality, calculated at a global or local scale, as well as under mixed or individual transportation mode (e.g., driving, biking and walking) with its directional accessibility, to uncover the structural properties of urban street networks. We further design an urban area transition network and apply PageRank to capture the intensity of human mobility. Our correlation analysis indicates different centrality metrics have different levels of correlation with the intensity of human movement. The closeness centrality consistently shows the highest correlation (with a coefficient around 0.6) with human movement intensity when calculated at a global scale, while straightness centrality often shows no correlation at the global scale or weaker correlation $\rho \approx 0.4$ at the local scale. The correlation levels further depend on the type of directional accessibility and of various types of transportation modes. Hence, the directionality and transportation mode, largely ignored in the analysis of road networks, are crucial. Furthermore, the strength of the correlation varies in the two cities examined, indicating potential differences in urban spatial structure and human mobility patterns.

**Keywords:** road network; centrality; urban mobility; transitions; geo-tagged tweets; correlation analysis

## 1. Introduction

Urban spaces are typically highly localized but they are globally connected [1]. In particular, the urban space consists of local patchworks, which serve some specific functionality. Nevertheless, these patchworks are linked by the urban street network into a whole at a global scale. The urban space expands to satisfy the requirement of human activities and extends based on the original urban structure to keep the whole structure flexible and sustainable. While the structure of urban space is greatly influenced by the history of each city [2], researchers have long been analyzing its properties in order to facilitate planning functionalities, such as resource allocation, transportation planning and help understand human movement patterns. The mobility activities collected in return on the urban space are then leveraged to analyze the evolution of urban space and help find hidden potentials for improvement [3].

Human activities in urban environments, such as business and travel, are often shaped and constrained by the geographical distance to and accessibility of the resources [4]. The

disparity of resource allocations across urban regions (e.g., moving from residential areas to shopping or art districts) or so-called intervening opportunities often motivate people to move across regions to satisfy their needs [5,6]. Urban mobility is also impacted by various other heterogeneous factors. For example, social interactions [7,8] also show strong correlations with human movements especially for distant travel. External events, e.g., global events like the COVID-19 pandemic [9–11], regional events like natural disasters [12], or local events like crime threats [13] and street festivals [14] could distort the regular patterns of urban mobility, of which the analysis can advocate the need for resilience and sustainability of economic activities in urban space. In this work, we focus on understanding the relationship between urban mobility and urban space structure captured by street networks.

The urban street network, functioning as the backbone of urban space, plays a vital role in connecting urban neighborhoods and supporting the local/global movement in/between urban areas. Its structural properties, such as centrality and accessibility, can reveal many implications on human activities. Centrality [15], which is a network-based metric measuring the structural importance of nodes in complex networks, is often utilized to capture the importance of different parts of road networks, such as intersections and segments. Intensive studies have shown road network centrality measures play important roles in understanding urban economic activities [1,16], exploring land use [17–20], traffic flow analysis [21–29], identifying traffic congestion [30] and discovering the patterns of traffic accidents [31]. For example, former studies [21,22] indicate that the structural properties of urban road networks as captured by the betweenness centrality can explain the observed traffic flow. Another form of centrality, closeness centrality, is shown to be highly correlated with the intensity of economic activities [16] and land use [32]. Furthermore, the aggregated human travel flow on streets is shown through simulations to be mainly shaped by the underlying street structure [23].

Previous studies often perform their analysis at the fine-grained street level. In this research, we instead consider an aggregated level with urban regions as study units, and in particular we explore the relationships between road network centrality measures and the intensity of human urban mobility. In particular, we study three different centrality measures, including closeness centrality, betweenness centrality, and straightness centrality. We explore the centrality calculations under various settings. At the global scale, we calculate the centrality by considering the whole road network; while at the local scale, we limit the calculation to neighboring road segments and intersections in a predefined spatial radius. We further consider different transportation modes (e.g., driving, biking, and walking) and their accessibility in both directions, which are often neglected in previous work while we will show later has a significant impact on our correlation analysis.

To capture human mobility, we collected large-scale geo-tagged social media posts as a proxy of real human movements. Compared to previous work on using survey-based and census data [33], Dollar bill tracking [34], Mobile Phone Call records [35], or trajectory data [36] collected from GPS (Global Positioning System) enabled devices, e.g., taxi trajectory, large-scale geo-tagged data from human sensors provide unprecedented opportunities for urban mobility analysis and modeling [37]. Geo-tagged social posts are often publicly available, relatively easier to collect, and cover movements with mixed modes of transportation.

Figure 1 presents the methodology workflow of our study. For road network centrality measure, we collected road network raw data from OpenStreetMap (www.openstreetmap.org), process it to get a list of nodes (i.e., road intersections) and edges (i.e., road segments), formulate the graph of road networks by whether or not considering the transportation modes the corresponding accessibility in both directions, and finally calculate various centrality measures. As a proxy of human urban mobility, we utilize geo-tagged tweets collected using Twitter Streaming API (https://developer.twitter.com/en/docs/tutorials/consuming-streaming-data). To capture the flow of the movements across spatial areas instead of static locations, we design an urban region transition network, where the intensity

of human movements between pairs of regions (i.e., the nodes in the network) is measured by the frequency of transitions. We further apply a personalized PageRank [38] to measure and rank the intensity of human mobility activities for each urban region. Finally, the relationship between road network structure and human urban mobility intensity are captured by a ranking correlation analysis between the centrality measures and Pagerank outputs.

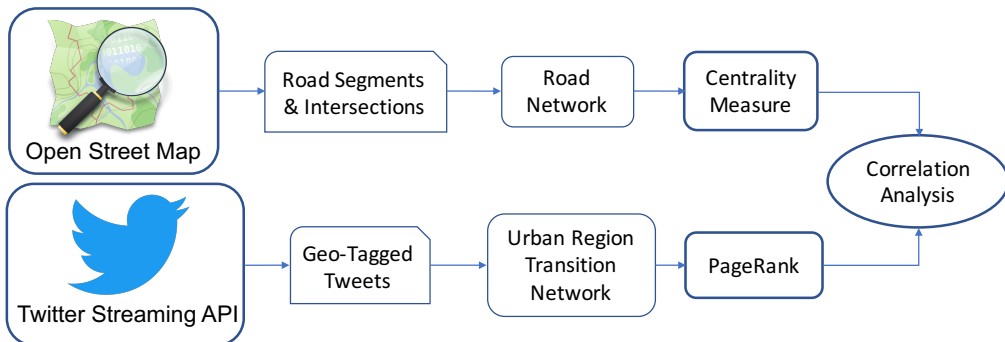

**Figure 1.** The methodology framework of our study. For road network centrality, we collected data from OpenStreetMap, parse to get road segments and intersections, formulate a graph (undirected or directed) by taking road segments as edges and intersections as nodes, and calculate various centrality measures. For human urban mobility intensity, we collected geo-tagged tweets as a proxy of human movements, designed an urban region transition network to capture the human movement flow across urban areas, and then applied a personalized PageRank algorithm to measure the intensity of urban mobility. Finally, we conducted a ranking correlation analysis between the two.

Our results imply that different centrality metrics correlate with the intensity of human movement at different levels. The directionality and transportation modes of urban road network do play an important role. Finally, the correlation strength further differs in the two cities examined. We summarize our main findings below

- With centrality measure calculated as global scale, closeness centrality which captures the accessibility has the highest correlations ($\rho \approx 0.6$) with the intensity of human urban mobility. Straightness centrality ranks second. Betweenness centrality often does not show a significant positive or occasionally shows slightly negative correlations.
- When calculated at a local scale with only nearby neighboring nodes considered, all centrality measures do not show significant positive correlations with urban mobility, except that the straightness centrality for the city Pittsburgh only shows a relatively weak correlation ($\rho \approx 0.3 - 0.4$).
- The transportation modes (i.e., driving, biking, and walking) with directional accessibility correlate with urban mobility at different levels. The centrality when considering the biking or walking mode tends to have higher correlations compared to the driving mode.
- The city Pittsburgh often shows stronger correlations than New York City, which could indicate the possible differences in terms of urban spatial structure and the routine travel and transportation modes of human movements. e.g., in New York City, the subway is the top choice for commute which cannot be captured by street networks.

**Roadmap:** The remaining of the paper is organized as follows: Section 2 discusses the related work and how our work differs. Section 3 describes our analysis setup and the dataset. Section 4 presents and analyzes our experimental results, while Section 5 concludes by also discussing the limitations and future work.

## 2. Related Work

In this section, we discuss previous works on road network structure analysis using centrality measure, and the various uses of road network centrality to help analyze human economic activities, including land use and urban mobility.

**Road Network Centrality Measure**: Centrality measure is originally borrowed from network science research and mainly applied in social networks, biological networks, and many others [15]. The measure has been extended to analyze the topological structure of the street network [39,40]. Work in [39] provides a comprehensive study on the statistical distribution of road network centrality measures, where closeness, betweenness and straightness centrality show very similar functional distributions, while some other centrality measures (e.g., Information centrality) follow a power-law distribution. They further show the distributions diversify across cities. To scale to a large road network, an approximation method for betweenness centrality was proposed in [41]. The work in [42] further proposed a method based on the shapley value to refine the existing centrality metrics such that they take into account not only the functioning of nodes as individual entities but also as members of groups of nodes. The original centrality algorithm can also need adjustments for certain applications, e.g., road network selection for maps at different scales [43] and explaining traffic flow [21]. In this work, we apply three standard centrality algorithms to capture the nodes (i.e., intersections) importance and study their relationships to urban mobility intensity.

**Road Network Centrality and Economic Activities**: Previous studies have been leveraging road network structure analysis to understand the urban form and spatial patterns [1] and how it relates to helping understand human economic activities. The work in [44] developed a model based on road network closeness centrality and residential land usage ratio to predict the population density, which achieves an acceptable accuracy. An exploratory study in [20] found road network betweenness centrality had the highest correlation with land-use intensity, second by closeness centrality. The former indicated the location advantage of being traversed more frequently plays an important role. Prior to the study, analysis done in different urban cities [17–19] had shown a strong association of road network structure capture centrality measures with land use. Work in [16] studied the correlation between street centrality and the density of economic activities, and they found that the street network centrality is more highly correlated with the density of secondary economic activities, like small local businesses, than primary economic activities, like wholesale. Studies also found the centrality of urban street network centrality has a high correlation with real estate values [32,45], gasoline prices [46] and spatial distribution and volume of retail stores [47,48]. For example, a study in [47] shows the spatial patterns of retail stores in road network structure. They found the Kernel Density Estimation of different types of retail stores (e.g., Restaurants) have significant levels of correlations with closeness centrality.

**Road Network Centrality and Urban Mobility**: Road network as the backbone connects the urban space and partially determines the accessibility and reachability of human movements. Intensive studies [21–29] found the road network structure captured by various centrality measures have a significant impact on urban traffic flow. Leung, Ian XY, et al. [22] utilized real-world GPS (Global Positioning System) traces data in Shanghai and San Francisco to prove that some modified centrality metrics can better predict traffic flow, and the power of prediction of street network centrality differed depending on the structural properties of street networks. A closer work was done by Jiang, Bin, et al. [23]. They simulated human movement and found that the aggregate flow on streets is mainly shaped by underlying street structure but not human traveling behavior, and closeness centrality is not a good indicator. However, they did not utilize real human movement data to verify the results. Work in [24] utilized taxi trajectory data and measure the traffic flow intensity as various spatial granularities, e.g., point, line, and area, and they found that the traffic flow intensity captured by weighted PageRank had a strong correlation with traffic at the line and area levels. Our work is similar to this work, while we instead utilize geo-tagged social media posts to capture traffic flow given the availability of GPS trajectory data. Work by [21] further proposed a modified betweenness centrality to better explain traffic flow. Other interesting lines of work related to traffic analysis focus on how road network centrality analysis helps routing and navigation [29], identifying traffic congestion

on road networks [30], discovering bottlenecks in road networks [49,50], investigating the patterns of traffic accidents [31] and evaluating accidents' impact on urban traffic mobility [25,51]. These works often take GPS trajectory data which are often limited in a specific city at a relatively small scale and limited to a single mode of transportation (e.g., taxi trajectory). Our work focuses on urban traffic flow across geographical regions, and we leverage geo-tagged social media data which are often at large-scale across urban cities with mixed transportation modes covered. Existing work also largely ignores the accessibility and directionality under certain transportation modes of street networks, which as we show later has a significant impact on the correlation between centrality measures and the intensity of urban mobility.

## 3. Experimental Setup and Datasets

In this section, we will introduce the network structures that capture the intensity of human movements and the urban road network as well as the data that drive their realizations in Pittsburgh and New York City (NYC).

### 3.1. Human Transition Network

In the human transition network $G_T = (U, E)$, the set of nodes $U$ is a collection of non-overlapping areas/neighborhoods in the city under examination. Further, a directed edge $e_{ij}$ between two areas $u_1, u_2 \in U$ exists if there has been observed a transition by a city-dweller from $u_1$ to $u_2$. The definition of $u_i$s can be arbitrary (e.g., municipal neighborhood borders, grids [52], areas divided by arterial roads [53], etc.). In our analysis, by simplicity, we follow the grid-based method [52] and divide the whole city ($10^2$ miles rectangle area considered around the center of each city) into 400 neighborhood areas, each one of 0.5 miles$^2$. The grid size of 0.5 miles$^2$ often covers 4–8 street blocks of economic districts in urban cities of the United States (US). We experimented with different grid sizes (i.e., 0.25, 0.75 miles) and our results are not sensitive to it. Further reducing the grid size leads to sparse data where many grids have 0 or very few geo-tweets and thus noisy for the latter statistical correlation analysis, while further increasing the grid size leads to a smaller number of grids which thus do not give enough data points for the latter analysis. We will also leave the exploration of other definitions of urban areas as future work.

In order to obtain the structure of $G_T$ for both cities we use geo-tagged social-media user-generated content. In particular, we use a dataset (https://www.icwsm.org/2016/datasets/datasets/, accessed on 31 October 2022) provided by [54], where the tweets were collected using Twitter's streaming API. Twitter's Terms of Service do not allow the full JSON for datasets of tweets to be distributed to third parties. However, they do allow datasets of tweet IDs to be shared. The provided dataset only contains Tweet ID, we further apply the tool "Hydrator" (https://github.com/DocNow/hydrator, accessed on 31 October 2022) to collect the original tweet post.

We consider tweets in a period from 15 July to 15 November 2013, and only keep those with exact geo-locations (i.e., with latitude and longitude) voluntarily shared by the user. Each processed tweet has a tuple format *<user Id, place Id, time, latitude, longitude>*. In total, we have 492,131 geo-tagged tweets in Pittsburgh and 3,172,872 in NYC. Figure 2 presents a scatter plot of the distribution, where we can see the central business areas, e.g., Pittsburgh downtown, and NYC Manhattan, often show a dense cluster of human movements. Using these data, we generate edge (transition) $e_{ij} \in E$ if the same Twitter user has generated two consecutive tweets in locations $l_i \in u_i$ and $l_j \in u_j$ within a predefined time interval $\Delta_t$ and the distance between these two locations is greater than a threshold $\Delta_d$. In our experiment, we set $\Delta_t = 4$ h and $\Delta_d = 10$ m. Figure 3 shows the Empirical Cumulative Distribution Function (ECDF) of the transition distance and time intervals. The ECDF is a step function defined as follows given $n$ data points $X_i$

$$\hat{F}_n(t) = \frac{1}{n} \sum_{i=1}^{n} \mathbb{1}_{X_i < t} \tag{1}$$

where $\mathbb{1}_{X_i < t}$ is an indicator function and equal to 1 when the $i_{th}$ data point $X_i$ is less than a fixed value $t$. $t$ is a sorted sequence of $X_i$ in increasing order. As we can see from the distribution, the parameters we select capture a majority of the transitions (e.g., >75%). Finally, we have 188,433 such transition pairs in Pittsburgh and 962,319 in NYC, as summarized in Table 1. Note that the above definition allows for self-edges in $G_T$. We can also annotate every edge $e_{ij}$ with a weight, which captures the number of transitions between the two urban areas $i$ and $j$.

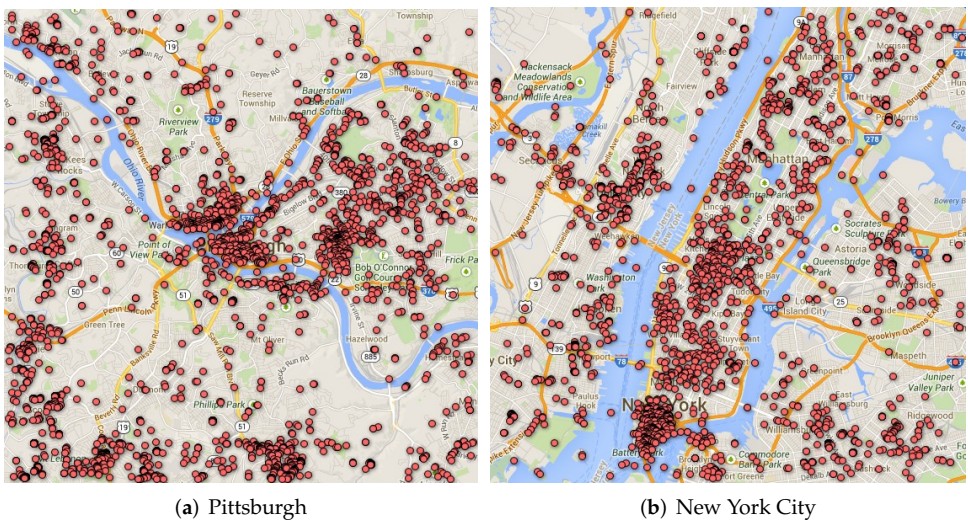

(**a**) Pittsburgh     (**b**) New York City

**Figure 2.** Distribution of geo-tagged tweets in different areas of the two cities, Pittsburgh (**a**) and New York City (**b**). The maps come from Google Maps with a zoom level of 10, and a scale ratio of 1:288895 given that Vector tile layers are used. For the map orientation, the north arrow follows naturally with the bottom-up direction in the figure. Each square image represents a 10 miles by 10 miles geographical region.

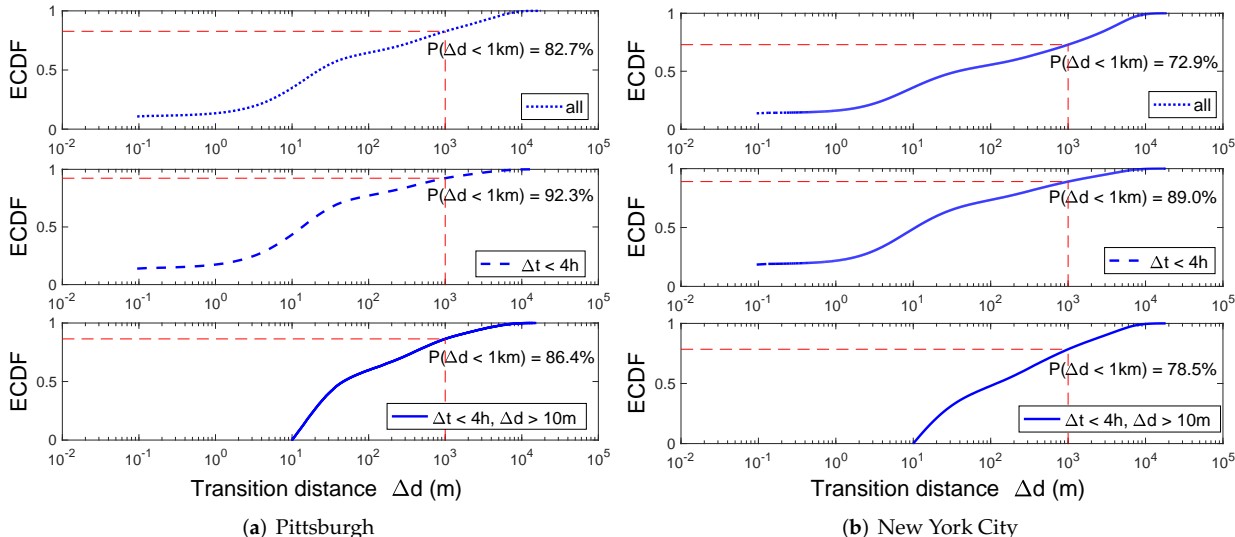

(**a**) Pittsburgh     (**b**) New York City

**Figure 3.** The Empirical Cumulative Distribution Function of transition distance under different parameters. The top figure includes all transitions, the one in the middle keeps transitions with time interval $\Delta t < 4$ h, while the bottom as the setting we select for our later analysis only keeps transitions with time interval $\Delta t < 4$ h and transition distance $\Delta d > 10$ m. Our parameter selection captures a majority of the transitions, i.e., 86.4% for Pittsburgh and 78.5% for New York City.

**Table 1.** Number of tweets and transitions in two urban cities, where a transition is a movement by the same user from one area to another area within a predefined time interval $\Delta_t$.

| City | # Geo-Tagged Tweets | # Transitions |
|---|---|---|
| Pittsburgh | 492,131 | 188,433 |
| New York | 3,172,872 | 962,319 |

**Centrality in $G_T$:** To capture the centrality of human movement in different neighborhoods, we calculate the PageRank [38] for each node in $G_T$. Originally, PageRank is used to capture the importance of web pages, where pages visited more often by a random walker are more important. In particular, we calculate a weighted PageRank score $P_i$ of a geographical area $i$ as:

$$P_i = \alpha \sum_j A_{ij} \frac{P_j}{k_j^{out}} + \beta_i \tag{2}$$

where, $\alpha = 0.85$ and $k_j^{out}$ is the weighted out-degree of node $j$ which counts self and outgoing edges. $\beta_i$ is a personalized (external) priority importance for area $i$, which is defined as the fraction of tweets taking place in area $i$. For implementation, we use a network analysis tool, namely "igraph", in R language (https://igraph.org/r/, accessed on 31 October 2022) to create the weighted graph and apply the existing PageRank algorithm available in igraph (https://igraph.org/r/doc/page_rank.html, accessed on 31 October 2022).

We will also use a second simple centrality metric for $G_T$, which is the number of geo-tagged tweets $n_{t,i}$ generated in area $i$. The latter does not incorporate mobility information, but rather captures the intensity of activity in each area.

### 3.2. Street Network

We will model the street network through a graph $G_s = (V, S)$, where the set of nodes $V$ represents the intersections in the street spatial structure and an edge $s_{ij} \in S$ represents the street segment that connects intersections $i$ and $j$. We fetch the street networks from OpenStreetMap and process them into the $G_s$ network format. In particular, we first exported the raw map data in 2014 for the Northeast region of the United States using the "Geofabrik Downloads" (https://download.geofabrik.de/, accessed on 31 October 2022) tool provided by OpenStreetMap. Then we apply `osm4routing` (https://github.com/Tristramg/osm4routing, accessed on 31 October 2022), which is a library originally written in Python and now rewritten in Rust, to extract the nodes and edges list to represent the road network.

Each node in the network (with a tuple *<node Id, latitude, longitude>*) represents the intersections of streets and the edge represents the segment. `osm4routing` extracts additional metadata such as the coordinates of each intersection, the length of each street segment and accessibility flags for each street segment in two directions (e.g., accessibility by car, foot, bike, etc.). The accessibility flags also provide a direction on each edge $s_{ij}$, which might be different depending on the mode of transportation.

The information of each edge is formalized as a tuple *<source node, target node, distance, car, car reverse, bike, bike reverse, foot>*, where the *distance* is the geographical distance between two nodes. *car* is an integer, for example, 2 means there are 2 lanes available for vehicles in the direction from the source node to the target node, and *car reverse* represents the capacity for the other direction. *Bike* and *foot* represent the accessibility for biking and walking separately. We select two cities, Pittsburgh and New York City, for our experiments, each with a 10 miles² rectangle area centered in the city. As summarized in Table 2, there are in total 23,126 (21,886) nodes and 32,475 (34,651) edges for the **undirected** $G_s$ in the $10^2$ miles rectangle area around the center of Pittsburgh (NYC). Figure 4 further visualizes the road network on the map by using the tool from [55]. In contrast to tweets' geographical distribution as shown in Figure 2, it is not quite straightforward to understand the relationship between the road network structure and urban mobility intensity. Note that for the road

network centrality measure below, we extend the studied area to 15 miles when calculating the node centrality to eliminate the "edge effects" [56] due to artificial boundaries.

**Table 2.** Number of nodes and edges for road networks in Pittsburgh and New York City.

| City | # Nodes | # Edges |
|---|---|---|
| Pittsburgh | 23,126 | 32,475 |
| New York | 21,886 | 34,651 |

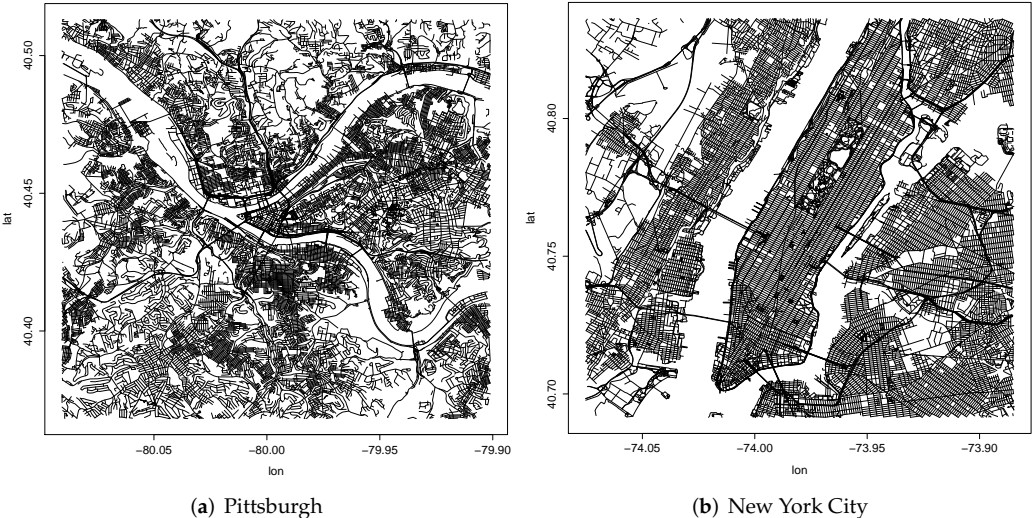

(**a**) Pittsburgh        (**b**) New York City

**Figure 4.** Street network in selected urban areas of two cities, Pittsburgh (**a**) and New York City (**b**). The map data come from OpenStreetMap, with the zoom level, scale ratio and north arrow orientation the same as in Figure 2.

**Centrality of Street Network:** For a road network $G_s$ with $n$ nodes and $m$ edges, we use a network analysis tool, namely "igraph", in R language (https://igraph.org/r/, accessed on 31 October 2022) to create the weighted graph and then calculate three well-established measures of node centrality: closeness centrality $C^c$, betweenness centrality $C^b$ and straightness centrality $C^s$.

$C_i^c$ captures the accessibility of node $i$ and is defined as [15]:

$$C_i^c = \frac{n-1}{\sum\limits_{j=1, j \neq i}^{n} d_{ij}} \tag{3}$$

where, $d_{ij}$ is the shortest path length between nodes $i$ and $j$.

$C_i^b$ quantifies to what extent node $i$ serves as a "broker" between nodes, is formally defined as [15]:

$$C_i^b = \frac{1}{(n-1)(n-2)} \sum_{s=1; t=1; s \neq t \neq i}^{n} \frac{n_{st}^i}{n_{st}} \tag{4}$$

where, $n_{st}$ is the number of shortest paths between nodes $s$ and $t$, while $n_{st}^i$ is the number of such shortest paths that traverse node $i$.

$C_i^s$ measures the extent to which node $i$ can be reached directly, on a straight line, from all other nodes, which is defined as [16]:

$$C_i^s = \frac{1}{n-1} \sum_{j=1; j \neq i}^{n} \frac{d_{ij}^{Eucl}}{d_{ij}} \tag{5}$$

where, $d_{ij}^{Eucl}$ is the Euclidean distance between nodes $i$ and $j$.

In particular, we calculate three global and nine local indices of street centralities. The global indices, $C_{glob}^c$, $C_{glob}^b$ and $C_{glob}^s$, are calculated using the whole road network. We also consider the local version of centralities $C_{local,d}^c$, $C_{local,d}^b$ and $C_{local,d}^s$, where we compute the centrality of node $i$ considering only the nodes that are within a radius $d$. In our experiments we use $d = 800$, 1600 and 2400 m, by only considering neighboring nodes within radius $d$.

To understand the relationship between different centrality measures, we calculate the pair-wise ranking correlations. Figure 5 shows the ranking correlations between pair-wise centrality measures when considering the street network as an undirected graph, for the two cities of interest (i.e., Pittsburgh and New York City). From the correlation color map, we can see the centrality measure at a global scale often have lower correlations with the one at local scale when $d$ gets smaller. At the global scale, the closeness centrality highly correlates with betweenness centrality but not with straightness centrality. However, straightness centrality at the local scale, especially when $d = 1600$, tends to correlate significantly higher with closeness and betweenness centrality at the global scale. Finally, given the same centrality measure at the local scale, e.g., $C_{local}^s$, the pairwise correlation between two local versions drops as the radius $d$ "difference" $\Delta d$ increase, e.g., $\rho(C_{local,d=800}^s, C_{local,d=1600}^s) = 0.86$ with $\Delta d = 800$ is larger than $\rho(C_{local,d=800}^s, C_{local,d=2400}^s) = 0.75$ with $\Delta d = 1600$, which is expected. However, given the same $\Delta d$, the larger the radius, the larger the pairwise correlation since the road networks between two local versions for the same node have a larger overlap ratio, e.g., the overlap ratio is 1/9 between $d = 800$ and $d = 1600$ versus 9/25 between $d = 1600$ and $d = 2400$.

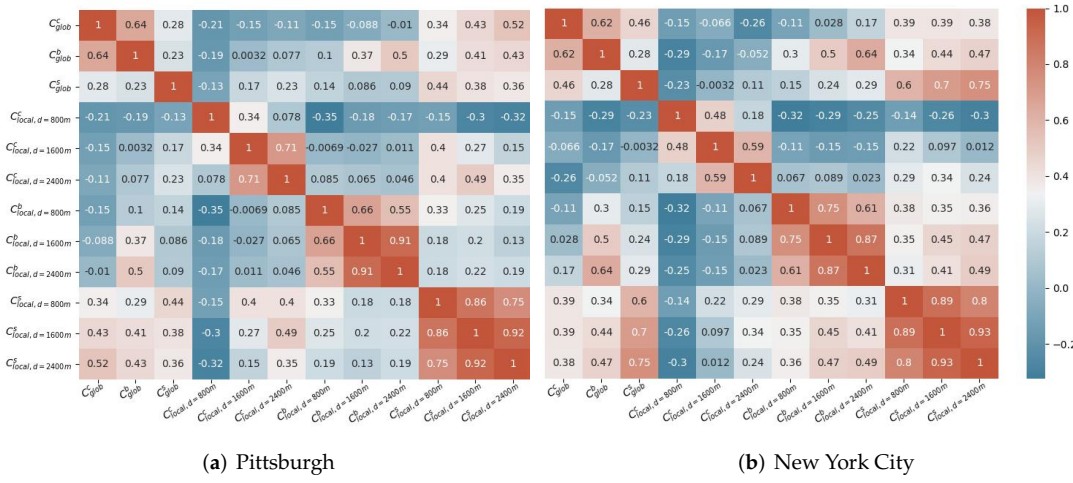

(a) Pittsburgh
(b) New York City

**Figure 5.** Pair-wise correlation between different centrality measures when considering the road network as an undirected graph. The color bar on the right side shows the range of correlation coefficients with the highest as 1 and the lowest as $-0.35$ per our data.

Finally, we consider the urban street network as a directed graph based on the direction accessibility for three types of movements including driving, biking and walking. In this case, there are two different calculations for closeness and straightness centrality based on two types of shortest paths between nodes. The first one is the outgoing shortest path $d_{ij}^{out}$, with the direction starting from node $i$ to node $j$. The second is the incoming shortest path $d_{ij}^{in}$ with direction into node $i$ from node $j$, capturing how easily a traveler can access node $i$ from other locations in the city. Therefore, we have *in* and *out* closeness and straightness centrality based on these two types of shortest path calculations. Figures 6 and 7 present the pair-wise correlations, where we can see *in* and *out* centrality does highly correlate with each other. They also show different correlation patterns compared to the one when considering the road network as an undirected graph.

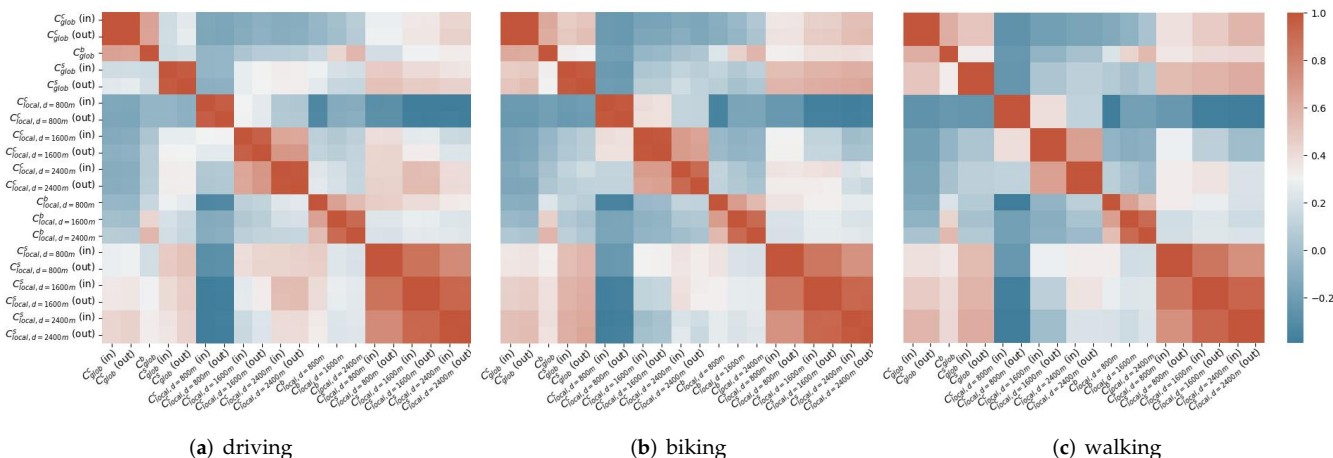

**Figure 6.** Pittsburgh: Pair-wise correlation between different centrality measures when considering the road network a directed graph. The color bar on the right side shows the range of correlation coefficients with the highest as 1 and the lowest as −0.4 per our data.

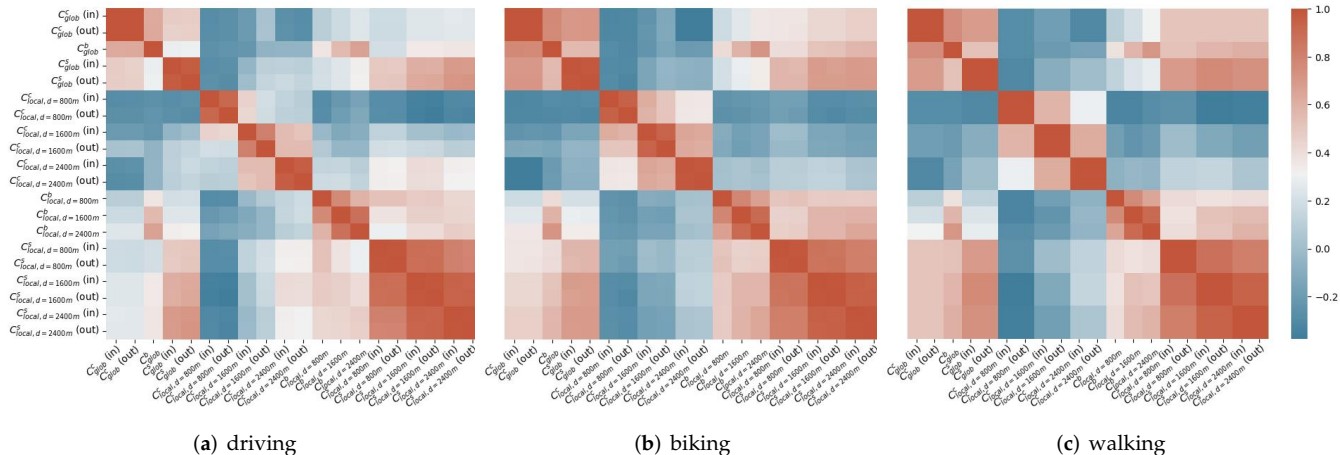

**Figure 7.** New York City: Pair-wise correlation between different centrality measures when considering the road network as a directed graph. The color bar on the right side shows the range of correlation coefficients with the highest as 1 and the lowest as −0.4 per our data.

### 3.3. Correlation Analysis Setup

Our goal is to examine the relationship between the central areas in a city as captured through the mobility of people, and the central areas of the city as captured through the street network. For that, we will utilize Spearman's rank correlation coefficient [57] $\rho$. In particular, the first variable for this correlation will be the PageRank centrality $P_i$ of nodes $i \in U$ (as well as $n_{t,i}$). However, the centrality values that we got from the street networks are defined on a different set of nodes (set $V$). Thus, we will use a spatial mapping $\Phi : V \to U$ utilizing the lat/lon coordinates we have for every $v \in V$. With $\Phi$ in place, the second variable for calculating $\rho$ will be the average road network centrality, $\bar{C}_v^*$, of all nodes $v \in V$ that map to $i \in U$, that is, $\Phi(v) = i$. we rank the 400 neighborhood areas based on average street centrality $\bar{C}_v^*$ and PageRank score $P$ of each area and get two sequences of ranking scores $r_C$ and $r_P$ and then calculate the Spearman's rank correlation coefficient $\rho$ to capture the level of correlations, which is defined as

$$\rho = \frac{\sum_i \left( r_{C_i} - \overline{r_C} \right) \left( r_{P_i} - \overline{r_P} \right)}{\sqrt{\sum_i \left( r_{C_i} - \overline{r_C} \right)^2 \sum_i \left( r_{P_i} - \overline{r_P} \right)^2}} \tag{6}$$

where, $r_{C_i}$ and $r_{P_i}$ are the ranking scores of $\bar{C}_i^*$ and $P_i$, separately. $\rho$ ranges from $-1$ to 1, where 1 is the total positive correlation and 0 is no correlation. For implementation, we used a statistical library "MASS" in R to calculate the ranks and their correlation.

## 4. Results and Analysis

We first take the urban street network as an undirected network without consideration of the traffic accessibility in two directions. Table 3 presents the correlation results for Pittsburgh and NYC. We can see that the global closeness centrality $C_{glob}^c$ and betweenness centrality $C_{glob}^b$ highly correlate with the intensity of human movement in both urban environments. This suggests that center areas in urban cities tend naturally to be more accessible from/to other places (higher $C_{glob}^c$) and thus function as city "hubs" (higher $C_{glob}^b$). In contrast, the global straightness centrality $C_{glob}^s$, local closeness centrality $C_{local,d}^c$ and local betweenness centrality $C_{local,d}^b$ present no significant positive correlations. However, the local straightness centrality $C_{local,d}^s$ shows an interesting urban difference with a significant level of correlation in Pittsburgh but not in NYC. This is more likely due to the difference of urban space structures or travel patterns between the two cities. Further analysis is needed to sort out the exact source of this difference.

In general, the level of correlations with urban mobility intensity using geo-tagged tweets align well with findings in previous studies using GPS trajectory data [22,24], while there are still some gaps in utilizing the road network structure to fully explain the urban traffic. This could be attributed to the mixed and complex transportation systems in urban areas [58], e.g., subway and train systems overlying the road network serve as "shortcuts" to connect urban regions and make them more accessible.

**Table 3.** Correlation $\rho$ (* indicates a $p$-value $< 0.05$; ** indicates $p$-value $< 0.01$) between the street centrality and the intensity of human movement.

| $G_T$ / $C$ | Pittsburgh | | NYC | |
|---|---|---|---|---|
| | $n_{t,i}$ | $P_i$ | $n_{t,i}$ | $P_i$ |
| $C_{glob}^c$ | 0.610 ** | 0.604 ** | 0.509 ** | 0.505 ** |
| $C_{glob}^b$ | 0.501 ** | 0.497 ** | 0.459 ** | 0.466 ** |
| $C_{glob}^s$ | 0.021 | 0.020 | 0.078 | 0.074 |
| $C_{local,d=800m}^c$ | $-0.223$ ** | $-0.228$ ** | $-0.085$ | $-0.093$ |
| $C_{local,d=1600m}^c$ | $-0.043$ | $-0.046$ | 0.012 | 0.004 |
| $C_{local,d=2400m}^c$ | 0.024 | 0.0189 | $-0.044$ | $-0.047$ |
| $C_{local,d=800m}^b$ | $-0.001$ | $-0.128$ * | 0.009 | $-0.127$ * |
| $C_{local,d=1600m}^b$ | 0.017 | 0.026 | $-0.072$ | $-0.070$ |
| $C_{local,d=2400m}^b$ | 0.106 * | 0.112 * | $-0.014$ | $-0.014$ |
| $C_{local,d=800m}^s$ | 0.348 ** | 0.351 ** | 0.105 * | 0.104 * |
| $C_{local,d=1600m}^s$ | 0.410 ** | 0.408** | 0.028 | 0.026 |
| $C_{local,d=2400m}^s$ | 0.442 ** | 0.438 ** | $-0.031$ | $-0.031$ |

We further present the results when we consider the urban street network as a directed graph based on the direction accessibility for three types of movements including driving, biking and walking. Table 4 presents the correlation between the centrality of directed street network and the PageRank score of neighborhood areas (results for $n_{i,t}$ are omitted due to space limitations). Compared to Table 3, we do not observe significant differences when considering the directed networks. This might be due to the fact that the transition network $G_T$ essentially captures the starting and ending point of a movement, ignoring the actual

path followed and/or due to the high similarity of the different directed network structures. Nevertheless, there is still some significant change for global straightness centrality when considering directed street networks—especially for biking and walking—which might be attributed to the fact that for these "slow modes" of transportation short geometric distance is important. Additionally, the gap between the two cities gets larger when considering the walking mode, which might indicate the different "walkability" of the urban space.

**Table 4.** Correlation results (* indicates a *p*-value $< 0.05$; ** indicates *p*-value $< 0.01$) by considering the road network as a directed network based on the accessibility of *driving*, *biking* and *walking* in either direction.

| PageRank<br>C | Driving | | Biking | | Walking | |
|---|---|---|---|---|---|---|
| | Pittsburgh | NYC | Pittsburgh | NYC | Pittsburgh | NYC |
| $C^c_{glob}$ (in) | 0.597 ** | 0.473 ** | 0.622 ** | 0.397 ** | 0.616 ** | 0.393 ** |
| $C^c_{glob}$ (out) | 0.594 ** | 0.481 ** | 0.623 ** | 0.391 ** | | |
| $C^b_{glob}$ | 0.481 ** | 0.431 ** | 0.520 ** | 0.452 ** | 0.514 ** | 0.444 ** |
| $C^s_{glob}$ (in) | −0.053 | 0.061 | 0.200 ** | 0.301 ** | 0.212 ** | 0.313 ** |
| $C^s_{glob}$ (out) | −0.002 | 0.083 | 0.231 ** | 0.303 ** | | |
| $C^c_{local,d=800m}$ (in) | −0.253 ** | −0.143 ** | −0.250 ** | −0.087 | −0.241 ** | 0.042 |
| $C^c_{local,d=800m}$ (out) | −0.282 ** | −0.142 ** | −0.253 ** | −0.069 | | |
| $C^c_{local,d=1600m}$ (in) | −0.133 ** | −0.170 * | −0.123 ** | −0.117 * | 0.103 * | −0.012 |
| $C^c_{local,d=1600m}$ (out) | −0.067 | 0.003 | −0.103 * | −0.012 | | |
| $C^c_{local,d=2400m}$ (in) | −0.053 | −0.215 ** | −0.024 | −0.178 ** | −0.011 | −0.078 |
| $C^c_{local,d=2400m}$ (out) | −0.039 | −0.204 ** | −0.077 | −0.171 ** | | |
| $C^b_{local,d=800m}$ | 0.042 | 0.100 * | 0.044 | −0.066 | 0.041 | −0.081 |
| $C^b_{local,d=1600m}$ | 0.061 | 0.125 * | 0.072 | −0.035 | 0.062 | −0.044 |
| $C^b_{local,d=2400m}$ | 0.140 ** | 0.100 * | 0.161 ** | 0.009 | 0.143 ** | 0.002 |
| $C^s_{local,d=800m}$ (in) | 0.248 ** | 0.053 | 0.324 ** | 0.002 | 0.362 ** | 0.094 |
| $C^s_{local,d=800m}$ (out) | 0.248 ** | 0.053 | 0.324 ** | 0.002 | | |
| $C^s_{local,d=1600m}$ (in) | 0.306 ** | 0.046 | 0.363 ** | −0.020 | 0.396 ** | 0.032 |
| $C^s_{local,d=1600m}$ (out) | 0.306 ** | 0.046 | 0.363 ** | −0.020 | | |
| $C^s_{local,d=2400m}$ (in) | 0.349 ** | 0.003 | 0.386 ** | −0.051 | 0.423 ** | −0.023 |
| $C^s_{local,d=2400m}$ (out) | 0.349 ** | 0.003 | 0.386 ** | −0.051 | | |

## 5. Conclusions and Discussions

In this paper, we examined the correlations between the centrality of street networks with the intensity of human movement in urban areas and we found that the correlation level differs with different centrality metrics, of which some further depend on different scales (global or local) of calculation, different cities, types of transportation modes and different directions.

We would like to emphasize that our analysis methods may suffer from a variety of biases. For example, we examine the correlation by aggregating the road network centrality and human movement in each neighborhood area, while a microscopic study might give a different view, e.g., traffic flow analysis at levels of individual street segment or intersection [21,22,59]. Also, the rectangle urban area we pick and grid-based region slicing may introduce edge effects on the correlation results. The alternatives would be

municipal neighborhood boundary or clustered regions based on spatial connectivity and human transitions [54]. Furthermore, the large-scale available dataset used here may have some noise and biases. For instance, the street networks in OpenStreetMap might not be that accurate especially for cities that are not that popular, since all the information is crowdsourced by the public. Also, the nature of voluntarily sharing may only give partial information of human movement captured by geo-tagged tweets, of which the quality depends on many other factors, such as demographic biases, spam tweets, and fake location information. Mixing of other data sources, e.g., bicycle sharing, GPS trajectory, subway records, or geo-tagged from other social media platforms could help eliminate such concerns.

Not shaded by the limitations, our work provides an illuminating way to study the relationship between urban structure and human movement in a large-scale way, given the public availability of road network data and geo-tagged social media posts. As we show closeness centrality often correlates the highest with human movement, but is still far from fully explaining it. A further smart combination of different centrality measures calculated at various settings is suggested to build stronger indicators for human urban mobility. We also recommend the consideration of the directional accessibility under different transportation modes enabled by road networks when performing road network structure analysis. Analysis performed in one city might not be easily transferable to another city, calling for a better understanding of the heterogeneity of spatial structure and mobility patterns across different cities.

In the future, we plan to examine the levels of correlation by considering the temporal and contextual information of human movements, such as the time and type [52]. For network centrality, we want to further investigate other practical factors, such as the max flow on a street (number of available lanes), the fastest path, and the density/type of resources surrounding a street intersection. We also want to trace the root cause of human movements, resource allocation, and road network convenience. Finally, we plan to understand the dynamic changes of road network structures and how they interplay with human urban mobility over time.

**Author Contributions:** Conceptualization, Li Geng and Ke Zhang; methodology, Li Geng and Ke Zhang; software, Ke Zhang; validation, Li Geng; formal analysis, Li Geng and Ke Zhang; investigation, Ke Zhang; resources, Ke Zhang; data curation, Ke Zhang; writing—original draft preparation, Ke Zhang; writing—review and editing, Li Geng; visualization, Li Geng. All authors have read and agreed to the published version of the manuscript.

**Funding:** This work was partly supported by PSC-CUNY grant #63160-00 51, funded by the Professional Staff Congress of the City University of New York.

**Data Availability Statement:** The geo-tagged tweets data used in this work is publicly available at https://www.icwsm.org/2016/datasets/datasets/. The last time we accessed the data was in May 2016.

**Conflicts of Interest:** The authors declare no conflict of interest. The authors declare no conflict of interest.

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
