# Peer review of "Correlation of Road Network Structure and Urban Mobility Intensity: An Exploratory Study Using Geo-Tagged Tweets"

_ijgi, doi:10.3390/ijgi12010007_

Round 1

Reviewer 1 Report

This study explored the relationships between street network centrality and the volume of people moving through urban areas, with different centrality metrics showing varying levels of correlation. The specific comments and suggestions for the Authors are:

In the abstract section, the author should include the quantitative results.

The introduction could be expanded, and more major research sources should be cited.

The use of centrality metrics in this research and their significance for urban mobility and transportation modes should be discussed by the author.

On page 3, The author should write the full form of GPS.

The author should mention the map scale and North Arrow in Figures 1 &3.

On page 5, the Author should Explain the ECDF in detail and write the full form.

The author should include the methodology framework graphically for this study for a better understanding of the flow of research work.

The author should mention the name of software’s/tools used for data analysis.

The author should add more logical justifications and suggestions in the conclusion section.

Author Response

Thanks for the valuable feedback and suggestions. Please see our aligned response in red with the corresponding revised parts in the revised paper.  All the revisions in the revised paper are highlighted in blue color. 

This study explored the relationships between street network centrality and the volume of people moving through urban areas, with different centrality metrics showing varying levels of correlation. The specific comments and suggestions for the Authors are:

In the abstract section, the author should include the quantitative results.

Please see the updated abstract with added representative quantitative results. 

The introduction could be expanded, and more major research sources should be cited.

We updated the introduction by adding discussions on previous work on urban mobility analysis (paragraph 2 in intro) and how road network centrality has been used to help with traffic flow analysis and others (paragraph 3), and how our work differs (paragraph 4-5). 

The use of centrality metrics in this research and their significance for urban mobility and transportation modes should be discussed by the author.

We expanded the introduction by adding paragraph 4 to indicate the importance of considering transportation modes and its accessibility in both directions for using centrality measures for urban mobility analysis.  

On page 3, The author should write the full form of GPS.

The author should mention the map scale and North Arrow in Figures 1 &3.

We added the full name of GPS when it is first shown in the paper. 

We updated the captions of both Figures (now they are Figure 2 & 4 in the revised paper) and describe the map source, scale and north arrow.  

On page 5, the Author should Explain the ECDF in detail and write the full form.

We added Equation 1 for ECDF, and explained it in details (see text in blue around Equation 1)

The author should include the methodology framework graphically for this study for a better understanding of the flow of research work.

We added Figure 1 in the introduction section for the method framework, and added a detailed description about the flow.  

The author should mention the name of software’s/tools used for data analysis.

We added the name and the source of the tools, e.g., igraph in R (line 222), MASS package in R for ranking correlation analysis (line 300), the tool to download the OpenStreetMap data (line 233), and the language of the tool we used to process OpenStreetMap raw data (line 233). 

The author should add more logical justifications and suggestions in the conclusion section.

We added a paragraph (line 357-365) in conclusion section to illustrate our findings with corresponding suggestions.  

Reviewer 2 Report

The manuscript has several grammatical errors and typos. 

It is necessary to revise the structure of the manuscript by better emphasizing the novelty of the research in the introductory part.

In addition, it is necessary to introduce more literature references explaining the evolution of urban spaces also taking into account the recent pandemic and the need for resilience and sustainability . In addition, there should be more explanation of how the evolution of GIS systems as well as Geo-Tagged Tweets systems can bring benefits to urban space planning and mobility 

Therefore, we recommend reading the following works 

1) Iranmanesh, A., & Alpar Atun, R. (2020). Reading the urban socio-spatial network through space syntax and geo-tagged Twitter data. Journal of Urban Design25(6), 738-757.

2) Tao, W. (2013). Interdisciplinary urban GIS for smart cities: advancements and opportunities. Geo-spatial Information Science16(1), 25-34.

3)  Alemdar, K. D., Kaya, Ö., Çodur, M. Y., Campisi, T., & Tesoriere, G. (2021). Accessibility of vaccination centers in COVID-19 outbreak control: A gis-based multi-criteria decision making approach. ISPRS International Journal of Geo-Information10(10), 708.

All acronyms should be written in expanded form when they are first included in the text 

It is necessary to include more commentary to Figure 2 by also increasing the font of the wording in the graphs 

It is necessary to include the sources of the maps used by including this in the caption of each figure or group of figures 

More commentary accompanying figure 4 and a more readable legend are needed

Limitations of the research as well as possible future steps of investigation should be emphasized in the concluding section.

Author Response

Thanks for your great feedback and suggestions. Please see our aligned response in red with the corresponding revised parts in the revised paper.  All the revisions in the revised paper are highlighted in blue color. 

The manuscript has several grammatical errors and typos. 

We ran multiple proofreading to correct the grammatical errors and typos we found. Please see those ones we replaced with our corrections which highlighted in blue.

It is necessary to revise the structure of the manuscript by better emphasizing the novelty of the research in the introductory part.

We restructure the introduction by adding more discussion on previous work, how our work differs with the novel parts. Please see paragraphs between line 63-80. 

In addition, it is necessary to introduce more literature references explaining the evolution of urban spaces also taking into account the recent pandemic and the need for resilience and sustainability . In addition, there should be more explanation of how the evolution of GIS systems as well as Geo-Tagged Tweets systems can bring benefits to urban space planning and mobility 

Therefore, we recommend reading the following works 

1) Iranmanesh, A., & Alpar Atun, R. (2020). Reading the urban socio-spatial network through space syntax and geo-tagged Twitter data. Journal of Urban Design, 25(6), 738-757.

2) Tao, W. (2013). Interdisciplinary urban GIS for smart cities: advancements and opportunities. Geo-spatial Information Science, 16(1), 25-34.

3)  Alemdar, K. D., Kaya, Ö., Çodur, M. Y., Campisi, T., & Tesoriere, G. (2021). Accessibility of vaccination centers in COVID-19 outbreak control: A gis-based multi-criteria decision making approach. ISPRS International Journal of Geo-Information, 10(10), 708.

We added these 3 suggested references and some other related ones, and discussed them accordingly in the introduction.  

All acronyms should be written in expanded form when they are first included in the text 

We added the expanded name for all acronyms when they first appear in the text, e.g., GPS, NYC, US, etc. 

It is necessary to include more commentary to Figure 2 by also increasing the font of the wording in the graphs 

We expanded the explanation in the caption of the Figure (now it is Figure 3 in the revised paper) and increased the font size to make it more visually readable. 

It is necessary to include the sources of the maps used by including this in the caption of each figure or group of figures 

We updated the captions for Figure 2 and 4 by mentioning the map source, e.g., Google Map and OpenStreetMap. 

Limitations of the research as well as possible future steps of investigation should be emphasized in the concluding section.

We added some discussion points and possible ways of investigations to address those limitations. Please see the text in blue in the 2nd paragraph of the conclusion section. 

Reviewer 3 Report

This paper focus on the relation between the road network structure and the urban mobility through the use of many different indicators and their correlations. The paper is mostly well written and quite clear about the test-cases used. The introduction is satisfactory. Some points should be clarified / expanded before publication.
- As stated in the conclusion, the correlations differ a lot with the centrality measures and give also different results for the two cities studied. What are the benefits of such a study (which indicators might be the best suited, can that be used to monitor changes through time, ...) ?
- The grid size used should play a major role due to the non-uniform repartition of the tweets in both cities. The authors said they experimented with different grid sizes but these were not very different from the original one. What happen with much smaller and wider grid sizes ? The authors undeline in the conclusion that a microscopic study might give a different view, this point should be expanded.
- Comments on Figure 4 should be expanded, and the fact that the correlation of the same indicators with greater distances does not follow a monotonic trend should be analyzed more thoroughly.

Author Response

Thanks for your great feedback and suggestions. Please see our aligned response in red with the corresponding revised parts in the revised paper.  All the revisions in the revised paper are highlighted in blue color. 

- As stated in the conclusion, the correlations differ a lot with the centrality measures and give also different results for the two cities studied. What are the benefits of such a study (which indicators might be the best suited, can that be used to monitor changes through time, ...) ?

We expanded the conclusion by talking about a few suggestions illuminated by our findings. Please see line 357-365 in the conclusion section.

- The grid size used should play a major role due to the non-uniform repartition of the tweets in both cities. The authors said they experimented with different grid sizes but these were not very different from the original one. What happen with much smaller and wider grid sizes ? The authors undeline in the conclusion that a microscopic study might give a different view, this point should be expanded.

The grid size of 0.5 miles often covers 4-8 street blocks of economic districts in urban cities of the United States (US). We experimented with different grid sizes (i.e., 0.25, 0.75 miles) but not others. Further reducing the grid size leads to sparse data where many grids have 0 or very few geo-tweets and thus noisy for the latter statistical correlation analysis, while further increasing the grid size leads to a smaller number of grids which thus not give enough data points for latter analysis. We will also leave the exploration of other definitions of urban areas as future work.

We updated the paper with such more explanations between lines 199-205. 

- Comments on Figure 4 should be expanded, and the fact that the correlation of the same indicators with greater distances does not follow a monotonic trend should be analyzed more thoroughly.

The figure shows the pairwise correlation of same indicators with greater distance “difference” shows a monotonic decreasing pattern. For example, for the same indicator, e.g., local straightness centrality, Clocal, d=800 has correlation coefficient = 0.86 with Clocal, d=1600 where Δd=800 , which it decreases to 0.75 with Clocal, d=2400, where Δd=1600.   This pattern also holds for the other two centrality indicators.   

We added more comments on this pair-wise correlation result (now it is Figure 5 in the revised paper). Please see more comments between lines 277-285.

Round 2

Reviewer 2 Report

the manuscript still has a few grammatical errors and typos. Once this is corrected the manuscript will be eligible for publication